# Human *RAP2A* homolog of the *Drosophila* asymmetric cell division regulator *Rap2l* targets the stemness of glioblastoma stem cells

Maribel Franco[1], Ricardo Gargini[2], Víctor M Barberá[3,4], Daniel Becerra[1], Miguel Saceda[3], Ana Carmena[1]*

[1]Instituto de Neurociencias, Consejo Superior de Investigaciones Científicas/ Universidad Miguel Hernández de Elche, Alicante, Spain; [2]Instituto de Investigación Biomédicas I+12 and Pathology and Neurooncology Unit, Hospital Universitario 12 de Octubre, Madrid, Spain; [3]Unidad de Investigación, Fundación para el Fomento de la Investigación Sanitaria y Biomédica de la Comunidad Valenciana (FISABIO), Hospital General Universitario de Elche, Elche, Spain; [4]Unidad de Genética Molecular, Hospital General Universitario de Elche, Elche, Spain

*For correspondence: acarmena@umh.es

Competing interest: The authors declare that no competing interests exist.

## eLife Assessment

This **useful** study explores the role of *RAP2A* in asymmetric cell division (ACD) regulation in glioblastoma stem cells (GSCs), drawing parallels to *Drosophila* ACD mechanisms and proposing that an imbalance toward symmetric divisions drives tumor progression. While findings on *RAP2A*'s role in GSC expansion are promising, and the reviewers found the study innovative and technically **solid**, the study relies on neurosphere models without in vivo confirmation and will therefore need to be further validated in the future.

**Abstract** Asymmetric cell division (ACD) is a fundamental process to balance cell proliferation and differentiation during development and in the adult. Cancer stem cells (CSCs), a very small but highly malignant population within many human tumors, are able to provide differentiated progeny by ACD that contribute to the intratumoral heterogeneity, as well as to proliferate without control by symmetric, self-renewing divisions. Thus, ACD dysregulation in CSCs could trigger cancer progression. Here, we consistently find low expression levels of *RAP2A*, the human homolog of the *Drosophila* ACD regulator *Rap2l*, in glioblastoma (GBM) patient samples, and observe that scarce levels of *RAP2A* are associated with poor clinical prognosis in GBM. Additionally, we show that restitution of RAP2A in GBM neurosphere cultures increases the ACD of glioblastoma stem cells (GSCs), decreasing their proliferation and expression of stem cell markers. Our results support that ACD failures in GSCs increase their spread and ACD amendment could contribute to reduce the expansion of GBM.

## Introduction

Asymmetric cell division (ACD) is a universal mechanism for generating cellular diversity during development, as well as for regulating tissue homeostasis in the adult (**Knoblich, 2008**). Stem and progenitor cells undergo ACD to simultaneously give rise to a self-renewing stem/progenitor cell

and to a daughter cell that is committed to enter a differentiation program. Remarkably, over the past years it has become apparent that undermining this critical process can lead to tumor-like overgrowth (*Knoblich, 2010*). This link between failures in the process of ACD and tumorigenesis was first established using the *Drosophila* neural stem cells, called neuroblasts (NBs), as a model system (*Caussinus and Gonzalez, 2005*). In this work, authors showed how *Drosophila* larval mutant brains for ACD regulatory genes were able to induce the formation of massive tumoral masses after weeks of being transplanted into the abdomen of wild-type fly hosts. Intriguingly, some genes firstly identified in *Drosophila* as tumor suppressor genes, such as *discs large 1* (*dlg1*), *lethal (2) giant larvae* (*l(2)gl*), and *brain tumor* (*brat*) (*Gateff, 1978*), were later uncovered as ACD regulators (*Albertson and Doe, 2003*; *Bello et al., 2006*; *Betschinger et al., 2006*; *Bowman et al., 2008*; *Ohshiro et al., 2000*; *Peng et al., 2000*; *Lee et al., 2006*), further supporting the connection between compromised ACD and tumorigenesis.

Tumor-initiating cells or cancer stem cells (CSCs) were originally identified in acute myeloid leukemia (AML) (*Bonnet and Dick, 1997*; *Lapidot et al., 1994*), but, to date, several studies have also revealed the existence of CSCs in multiple solid tumors (*Bajaj et al., 2020*). These CSCs represent a very small but highly malignant population within the tumor, able to proliferate without control by symmetrical, self-renewal divisions, as well as provide differentiated progeny by ACD that contribute to the heterogeneity of the tumor. These properties, along with their quiescent state, increase in metabolic activity and high capacity of DNA repair, make CSCs extremely resistant to radio- and chemotherapy and responsible for malignant relapse (*Bajaj et al., 2020*; *Reya et al., 2001*; *Moore and Lyle, 2011*; *Rosen and Jordan, 2009*; *Yadav et al., 2020*). Thus, understanding the unique nature of these cells and their landmarks is crucial to target CSCs and hence completely destroy the tumor. An intriguing possibility is that an imbalance in the mode of CSC divisions, favoring symmetric renewal divisions to the detriment of asymmetric divisions, might contribute to the transition from a chronic (low grade) to an acute (high grade) phase in the cancer progression (*Bajaj et al., 2015*; *Chao et al., 2024*).

Glioblastoma (GBM), the most aggressive and lethal brain tumor with very poor prognosis, is among the solid tumors in which the presence of CSCs, called glioma or glioblastoma stem cells (GSCs), has been proven. As other CSCs, GSCs are responsible for the formation, maintenance, resistance to conventional therapy, and consequent relapse of GBM (*Galli et al., 2004*; *Singh et al., 2003*; *Singh et al., 2004*; *Biserova et al., 2021*; *Chen et al., 2012*). Several GSC biomarkers, such as CD133 and Nestin, and signaling pathways, such as Notch and Hedgehog, have been identified to help isolate and target these crucial cells (*Bajaj et al., 2020*; *Hassn Mesrati et al., 2020*; *Tang et al., 2021*). In spite of that, there are still many unknowns regarding the nature, properties, and origin of CSCs, in general, and GSCs, in particular.

Here, we analyze the consequences of compromising ACD in the GSCs within GBM neurosphere cell cultures. Specifically, we show that *RAP2A*, the human homolog of *Rap2l*, which encodes a *Drosophila* small GTPase and novel ACD regulator, displays low expression levels in GBM and that restitution of RAP2A in GBM neurosphere cultures increases the ACD and decreases the expression of stem cell markers in the GSCs. Our results support that failures in ACD increase GSC proliferation, promoting their spreading, and that ACD amendment might contribute to reduce the expansion of GBM.

## Results
### *RAP2A* is highly downregulated in GBM patients
As a first approach to analyze whether ACD might be impaired in GBM, we analyzed the expression levels of human homologs of 21 *Drosophila* ACD regulators in a GBM microarray (*Larriba et al., 2024*; *Figure 1A*). We also investigated which of those genes were more similar in expression pattern to the others by performing a Gene Distance Matrix (GDM) (*Figure 1B*). The human genes *TRIM2* and *RAP2A* showed the lowest levels in the GBM patient samples analyzed. *TRIM2* belongs to the same family as *TRIM3* and *TRIM32*, all of them related to the *Drosophila* ACD regulator gene *brat*. *TRIM3*, the closest homolog of *brat*, has been previously found to be expressed at low levels in GBM (*Chen et al., 2014*). Hence, we concentrated on *RAP2A*, the second human gene consistently found at very low levels in all the patient samples (*Figure 1—figure supplement 1A*). *RAP2A* is also known to have a tumor suppressor role in the context of glioma migration and invasion (*Wang et al., 2014*; *Wang et al., 2017*). First, to further support the microarray data expression levels in a bigger

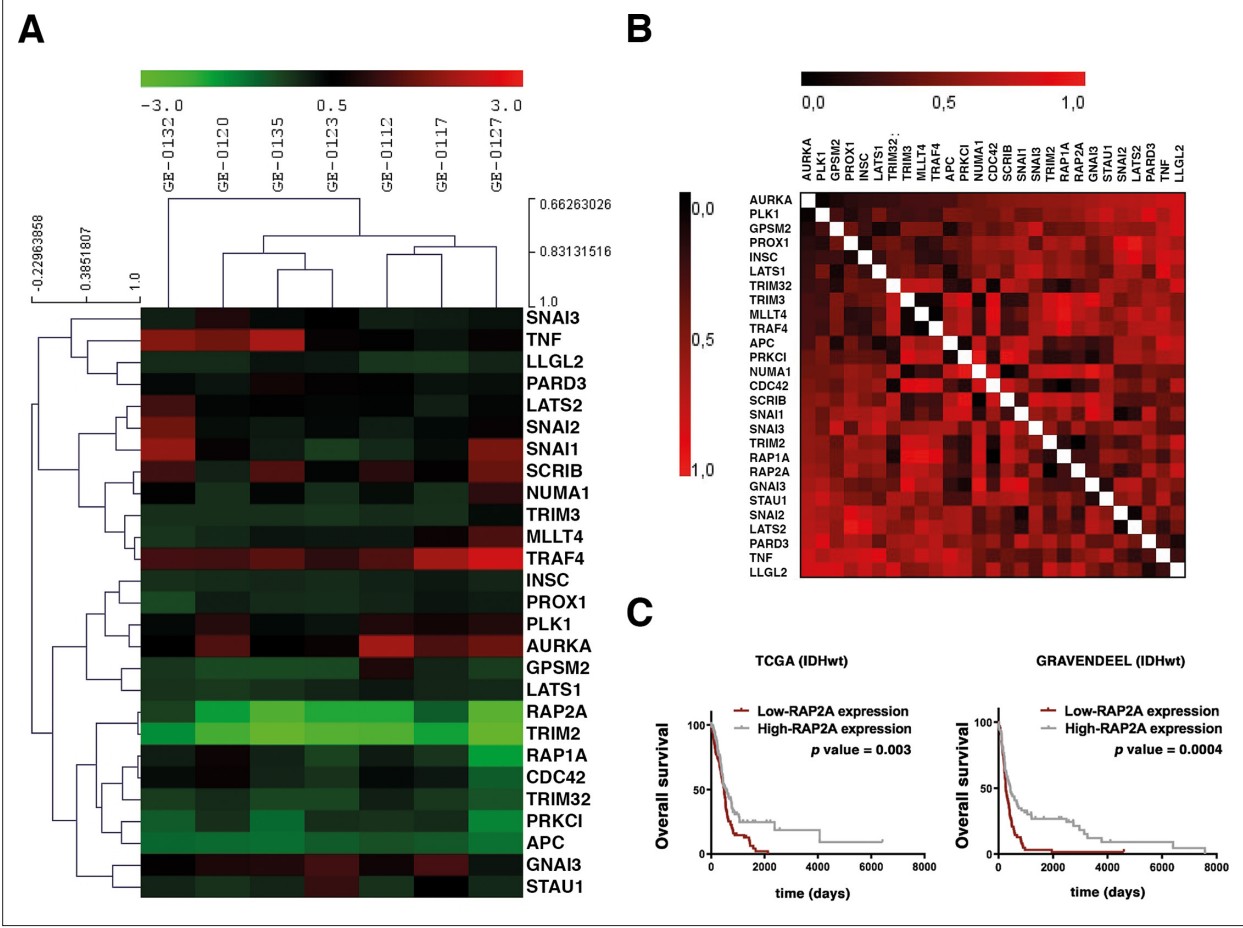

**Figure 1.** *RAP2A* is highly downregulated in glioblastoma (GBM) patients. (**A, B**) Analysis of GBM patient versus control samples focused on the level of expression of human homologs of *Drosophila* asymmetric cell division (ACD) regulators by hierarchical clustering (**A**) and the Gene Distance Matrix (**B**). Color-coded scale bar indicates the fold level of expression. (**C**) Kaplan–Meier survival curves corresponding to patients in The Cancer Genome Atlas (TCGA) (IDHwt) and Gravendeel (IDHwt) cohort in GBM datasets. Low *RAP2A* expression levels in human GBM are associated with a poor prognosis.

The online version of this article includes the following figure supplement(s) for figure 1:

**Figure supplement 1.** Statistical analysis of asymmetric cell division expression levels in glioblastoma (GBM) patient samples.

**Figure supplement 2.** RAP2A expression levels in the different GBM subtypes and their associated survival rates.

sample size, we performed in silico analyses taking advantage of established datasets of GBM IDH wild-type, such as 'The Cancer Genome Atlas' (TCGA) and 'Gravendeel' cohorts. We confirmed that low *RAP2A* expression levels in human GBM are associated with a poor prognosis (*Figure 1C*). In a multivariate analysis in the TCGA cohort, when we included the variable's RAP2A expression, patient age, and standard chemo-/radiotherapy treatment in the model, it was observed that RAP2A levels and patient age have a significant implication in patient survival as an independent variable. In fact, specific values were significant in predicting overall survival (OS). For example, 'RAP2A expression' with a hazard ratio (HR) of 0.866 (95% CI 0.760–0.986) and a p-value of 0.03, and 'patient age' with an HR of 1.030 (95% CI 1.022–1.037) and a p-value of <0.001. Conversely, standard chemo-/radiotherapy treatment did not remain as an independent variable, with an HR of 0.820 and a p-value of 0.053. We also analyzed RAP2A levels in the different GBM subtypes (proneural, mesenchymal, and classical) in the TCGA cohort and their prognostic relevance. The proneural subtype that displayed RAP2A levels significantly higher than the others was the subtype that also showed better prognosis (*Figure 1— figure supplement 2*). Hence, we focused on the human gene *RAP2A* to analyze its potential role in ACD and its relevance in GBM.

### Drosophila RAP2A homolog Rap2l regulates ACD

We previously showed that the *Drosophila* gene *Rap1* regulates the ACD process (*Carmena et al., 2011*). The closest human homolog of *Drosophila Rap1* is *RAP1A*, while human *RAP2A* is more similar to *Drosophila Rap2l*, which has not been previously analyzed in the context of ACD. Hence, we first wanted to investigate whether *Rap2l* was, like *Rap1*, implicated in ACD regulation. With that aim, we started analyzing the ACD of neural stem cells (NBs) and intermediate neural progenitors (INPs) within *Drosophila* larval brain type II NB lineages (NBII) (*Figure 2A*; *Bowman et al., 2008*; *Bello et al., 2008*; *Boone and Doe, 2008*). In NBII lineages, there is one NB that expresses the transcription factor Deadpan (Dpn) and several INPs, which express both Dpn and the transcription factor Asense (Ase) (*Figure 2A*). We first observed the presence of ectopic NBs (eNBs; Dpn$^+$ Ase$^-$) after downregulating *Rap2l* in NBII lineages in four out of five mutant brains analyzed (n=34 NB lineages), while in control lineages only one NB was usually found (*Figure 2B*). To further determine the specificity and penetrance of the phenotype, we repeated this experiment analyzing this *Rap2l* RNAi line (KK) along with two additional *Rap2l* RNAi lines (GD and BDSC), and substantially increasing the number of samples (both the number of NB lineages and the number of brains analyzed) (*Figure 2—figure supplement 1*). All the different lines showed a significant number of eNBs (*p<0.05, **p<0.01, or ***p<0.001; n>134 NB lineages analyzed) in all or almost all of the different brains analyzed (n>14) (see *Figure 2—figure supplement 1* for details). This result suggested that the process of ACD was impaired within those *Rap2l* mutant NBII lineages. To more directly support that, we analyzed the localization of key ACD regulators in metaphase NBs and INPs of NBII lineages, such as the apical proteins aPKC and Canoe (Cno), and the cell-fate determinant Numb (*Wodarz et al., 2000*; *Knoblich et al., 1995*; *Speicher et al., 2008*). In control metaphase NBs and INPs, those ACD regulators form crescents at the apical (aPKC and Cno) or basal (Numb) poles of the dividing cell (*Figure 2C*). The apical localization of aPKC was not significantly altered after downregulating *Rap2l* in NBII lineages; however, the localization of Cno and Numb did fail in most of the *Rap2l* mutant brains analyzed (in five out of six in the case of Cno, and in three out of four in the case of Numb) (*Figure 2C*). All these results strongly suggested that Rap2l, as Rap1, was an ACD regulator. Hence, we came back to the human homolog of *Drosophila Rap2l*, *RAP2A*, to determine its potential relevance in the context of the ACD of GSCs.

### RAP2A expression in GBM neurosphere cultures reduces the stem cell population

After confirming by in silico analyses that low *RAP2A* expression levels in GBM patients were associated with a poor prognosis, we decided to use established GBM cell lines to investigate in neurosphere cell cultures whether the downregulation of *RAP2A* was affecting the properties of GSCs. We tested six different GBM cell lines, finding similar mRNA levels of *RAP2A* in all of them and significantly lower levels than in control Astros (*Figure 3A*). We focused on the GBM cell line called GB5, which grew well in neurosphere cell culture conditions, for further analyses. Given that stem cell markers, such as CD133, SOX2, and NESTIN, are landmarks of GSCs, we started analyzing their expression after restoring RAP2A in the neurosphere culture (*Figure 3B*). Immunofluorescences of each of those markers revealed a significant reduction in the protein intensity in the RAP2A-expressing neurospheres (GB5-RAP2A) compared to the control neurospheres (GB5) (*Figure 3C*). Likewise, RT-PCRs and western blots also showed a significant decrease in the mRNA and protein expression levels, respectively, of those stem cell markers (*Figure 4*). Hence, RAP2A seems to be contributing to decrease the stem cell population within GBM neurosphere cultures.

### RAP2A expression in GBM neurosphere cultures decreases cell proliferation and sphere size

Next, we wondered whether the reduction in the stem cell population after expressing *RAP2A* in neurosphere cultures was reflected in the level of cell proliferation and, consequently, size of the neurospheres. To investigate this, we first analyzed the expression of the proliferation marker and key tool in cancer diagnostics Ki-67, highly expressed in cycling cells but absent in quiescent (G0) cells (*Gerdes et al., 1983*; *Sun and Kaufman, 2018*). We detected a significant decrease in the number of proliferating cells per neurosphere in those cultures expressing *RAP2A* (*Figure 5A*). Moreover, the average area size of these GB5-RAP2A neurospheres was also significantly reduced compared to control GB5 neurospheres (*Figure 5B*). Thus, RAP2A promotes a decrease in cell proliferation.

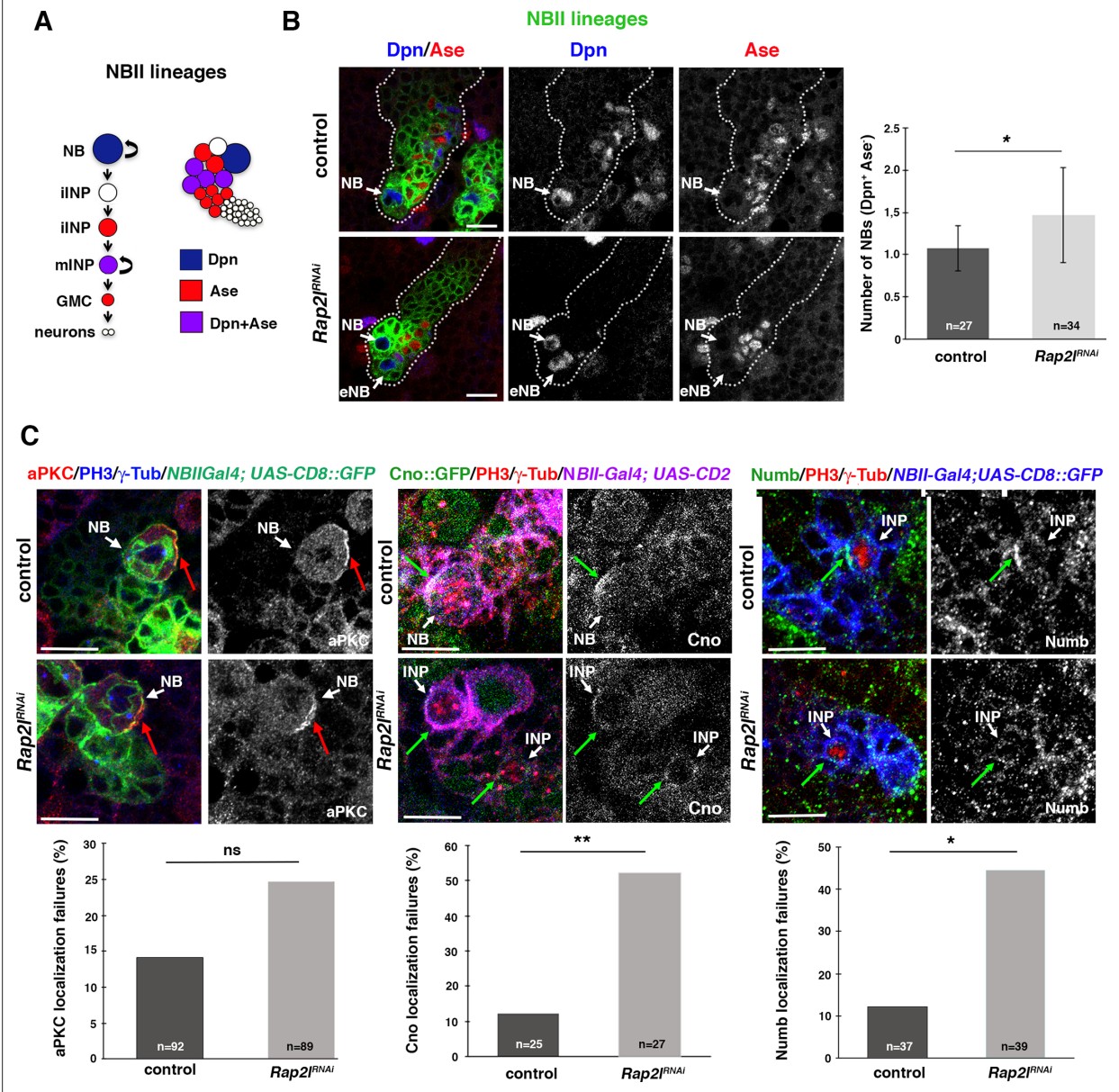

**Figure 2.** *Drosophila RAP2A* homolog *Rap2l* regulates asymmetric cell division (ACD). (**A**) *Drosophila* type II neuroblast (NBII) lineage; the only NB per lineage expresses the transcription factor Dpn, while mature intermediate neural progenitors (INPs) express both Dpn and Ase; GMC, ganglion mother cell; iINP, immature INP; mINP, mature INP. (**B**) Confocal micrographs showing a control larval brain NBII lineage with one NB (Dpn⁺ Ase⁻) and an NBII lineage in which *Rap2l* has been downregulated displaying an ectopic NB (eNB). (**C**) *Rap2l* downregulation in NBII lineages by the *wor-Gal4 ase-Gal80* (an NBII-specific driver) causes Cno and Numb localization failures (green arrows) in dividing progenitors (NB or INPs), while aPKC localization is not significantly altered (red arrows). Data shown in the scaled bar graphs was analyzed with a Mann–Whitney *U* test (**B**) or a chi-square test (**C**), *p<0.05, **p<0.01, ns, not significant, n=number of NB lineages analyzed (in **B**) or number of dividing cells analyzed (in **C**); scale bar: 10 μm.

The online version of this article includes the following source data and figure supplement(s) for figure 2:

**Source data 1.** Source data of *Figure 2B* analysis.

**Source data 2.** Source data of *Figure 2C* (aPKC) analysis.

**Source data 3.** Source data of *Figure 2C* (cno) analysis.

**Source data 4.** Source data of *Figure 2C* (Numb) analysis.

**Figure supplement 1.** *Drosophila RAP2A* homolog *Rap2l* regulates asymmetric cell division (ACD).

**Figure supplement 1—source data 1.** Source data of *Figure 2—figure supplement 1* analysis.

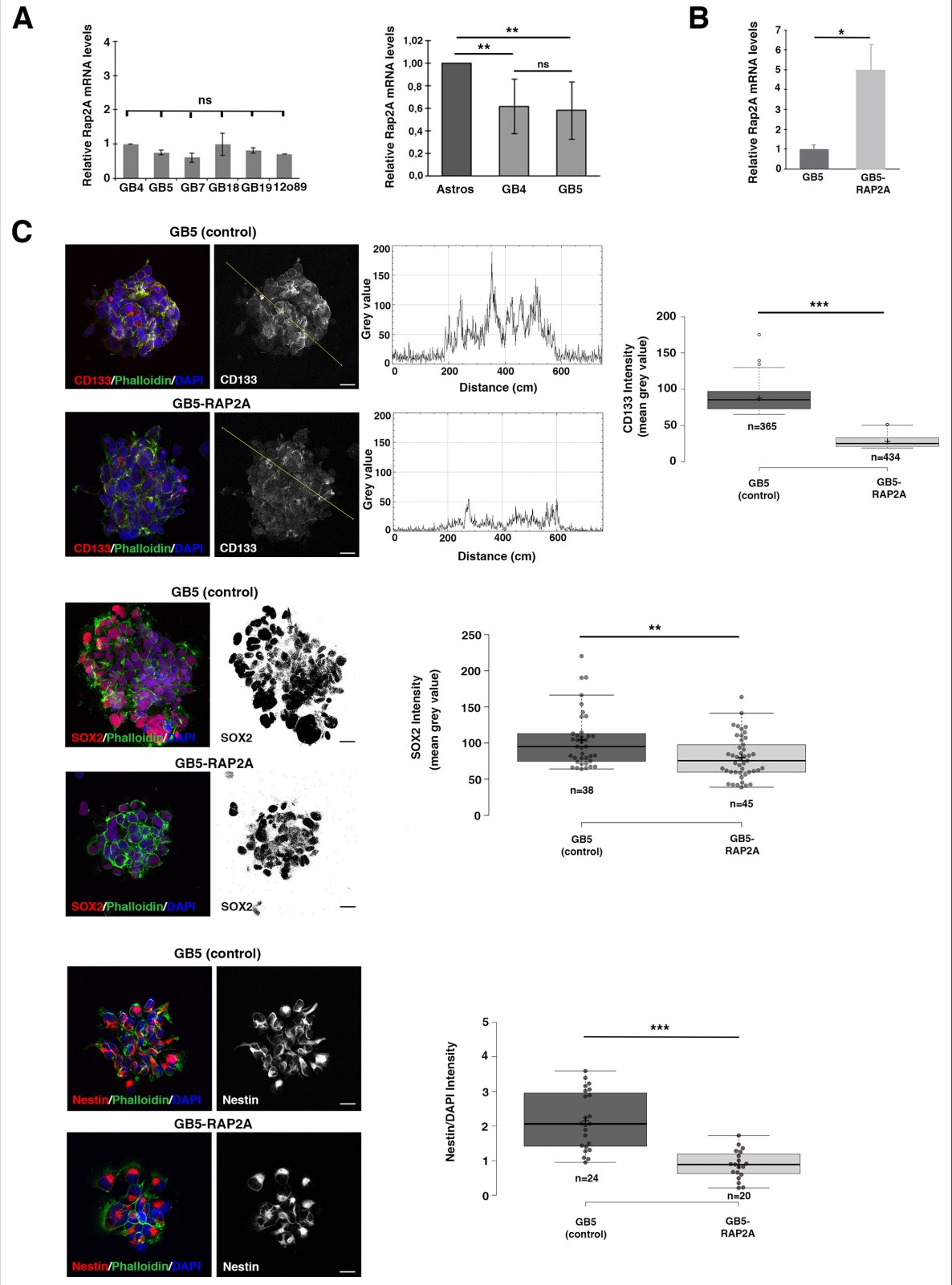

**Figure 3.** *RAP2A* expression in glioblastoma (GBM) neurosphere cultures reduces the stem cell population. (**A**) Different GBM cell lines show similar *RAP2A* mRNA levels and significantly lower levels than in control Astros. (**B**) *RAP2A* mRNA levels are significantly higher in the GB5 line after infecting this line with *RAP2A* (GB5-RAP2A). Data shown in the scaled bar graphs in (**A**) and (**B**) was analyzed with an ANOVA and a *t*-test, respectively; error bars show the SD; n=2 (in **A**) and 3 (in **B**) different experiments. (**C**) Immunofluorescences of the glioblastoma stem cell (GSC) stem cell markers CD133,

*Figure 3 continued on next page*

*Figure 3 continued*

SOX2, and Nestin reveal a significant reduction in the protein intensity in the GB5 RAP2A-expressing neurospheres (GB5-RAP2A) compared to the control neurospheres (GB5). Data shown in the box plots was analyzed with a Mann–Whitney *U* for CD133 and Nestin and with a *t*-test for Sox; the central lines represent the median and the box limits the lower and upper quartiles, as determined using R software; crosses represent sample means; error bars indicate the SEM; n=number of sample points; *p<0.05, **p<0.01, ***p<0.001, ns, not significant; scale bar: 10 μm.

The online version of this article includes the following source data for figure 3:

**Source data 1.** Source data of *Figure 3C* (CD133) analysis.

**Source data 2.** Source data of *Figure 3C* (SOX2) analysis.

**Source data 3.** Source data of *Figure 3C* (Nestin) analysis.

## *RAP2A* expression in GBM neurosphere cultures fosters ACD in GSCs

Given the reduction in cell proliferation and stem cell population after adding RAP2A to the control GB5 neurosphere cultures, we wanted to investigate whether RAP2A was favoring an asymmetric mode of cell division in these cells. Thus, we decided to follow over time the GB5-RAP2A and the control GB5 neurospheres, analyzing the number of cells per neurosphere. We reasoned that an even number of cells per neurosphere would be indicative of ACDs, while an odd number of cells would point to an asymmetric mode of cell division (see also 'Materials and methods' for details). We observed a significant increase in neurospheres with an odd number of cells in the GB5-RAP2A neurospheres (n=124 neurospheres analyzed) compared to control GB5 neurospheres (n=153 neurospheres) (*Figure 6A*). This result was very suggestive of an increase in the number of ACDs in GB5-RAP2A neurosphere cells. To further support this result, we decided to analyze the distribution of the cell fate determinant NUMB, a key conserved ACD regulator whose asymmetric distribution in the mother and progeny ultimately promotes the repression of stem cell self-renewal in the daughter cell in which it is segregated (*Yan, 2010*; *Morrison and Kimble, 2006*; *Cayouette and Raff, 2002*; *Cicalese et al., 2009*; *Ortega-Campos and García-Heredia, 2023*). We observed a significant increase in the number of GB5-RAP2A dividing cells that showed an asymmetric localization of NUMB in the progeny compared to control GB5 dividing cells (*Figure 6B*). Thus, all these results strongly supported a function of RAP2A promoting ACD in the GSCs.

## Discussion

ACD is an evolutionary conserved mechanism used by stem and progenitor cells to generate cell diversity during development and regulate tissue homeostasis in the adult. CSCs, present in many human tumors, can divide asymmetrically to generate intratumoral heterogeneity or symmetrically to expand the tumor by self-renewal. Over the past 15 years, it has been suggested that dysregulation of the balance between symmetric and ACDs in CSCs, favoring symmetric divisions, can trigger tumor progression in different types of cancer (*Bajaj et al., 2015*; *Chao et al., 2024*; *Li et al., 2022*), including mammary tumors (*Cicalese et al., 2009*; *Dey-Guha et al., 2011*), GBM (*Chen et al., 2014*), oligodendrogliomas (*Sugiarto et al., 2011*; *Daynac et al., 2018*), colorectal cancer (*Bu et al., 2013*; *Hwang et al., 2014*), and hepatocellular carcinoma (*Hwang et al., 2014*). Thus, it is of great relevance to get a deeper insight into the network of regulators that control ACD, as well as the mechanisms by which they operate in this key process.

*Drosophila* neural stem cells or NBs have been used as a paradigm for many decades to study ACD (*Homem and Knoblich, 2012*). NBs divide asymmetrically to give rise to another self-renewing NB and a daughter cell that will start a differentiation process. Over all these years, a complex network of ACD regulators that tightly modulate this process has been characterized. For example, the so-called cell-fate determinants, including the Notch inhibitor Numb, accumulate asymmetrically at the basal pole of mitotic NBs and are exclusively segregated to one daughter cell, promoting in this cell a differentiation process. The asymmetric distribution of cell-fate determinants in the NB is, in turn, regulated by an intricate group of proteins asymmetrically located at the apical pole of mitotic NBs, generically known as the 'apical complex'. This apical complex includes kinases (i.e., aPKC), small GTPases (i.e., Cdc42, Rap1), and Par proteins (i.e., Par-6, Par3), among others (*Homem and Knoblich, 2012*).

Given the potential relevance of ACD in CSCs, we decided to take advantage of all the knowledge accumulated in *Drosophila* about the network of modulators that control asymmetric NB division. As

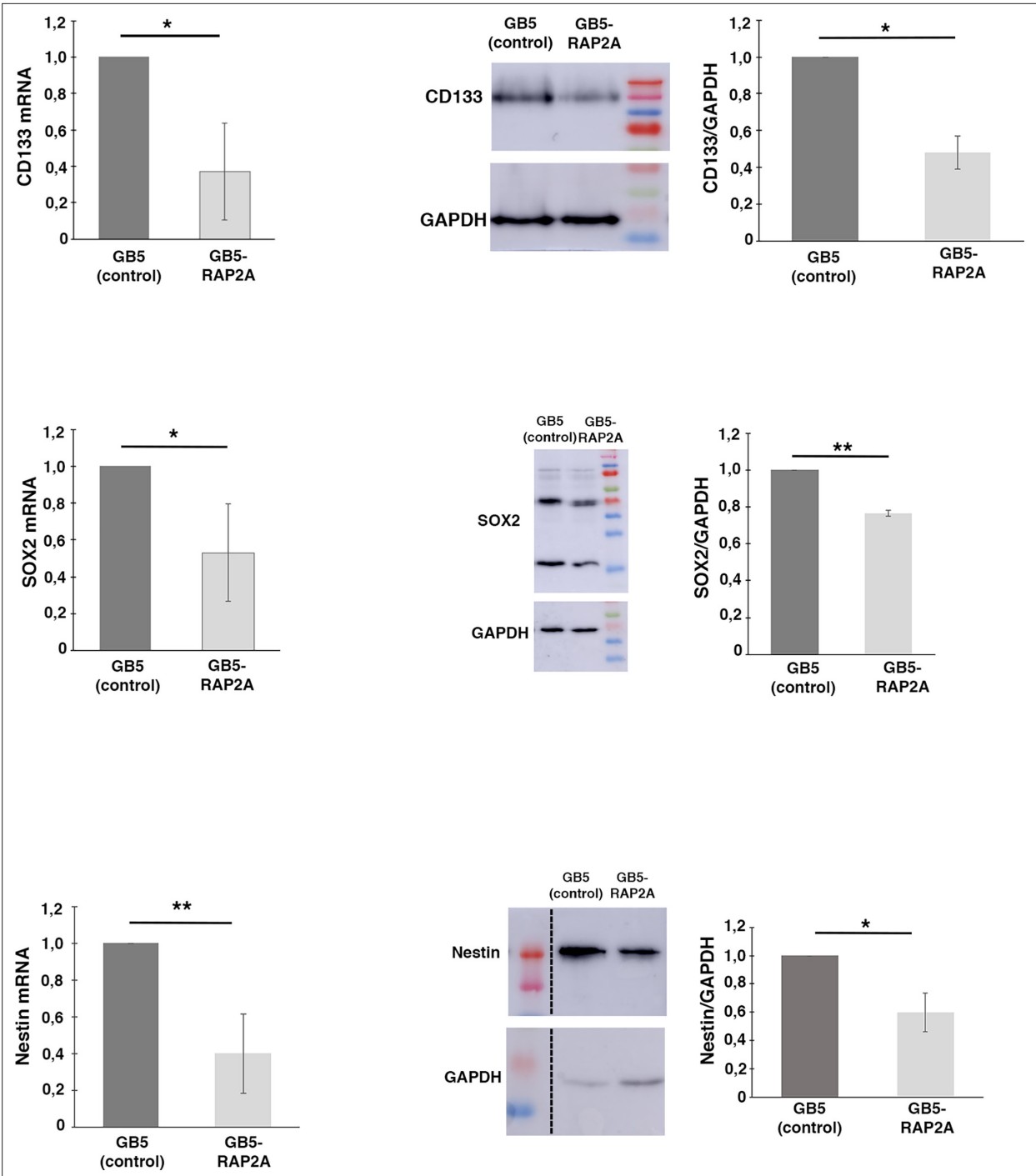

**Figure 4.** *RAP2A* expression in glioblastoma (GBM) neurosphere cultures reduces the stem cell population. RT-PCRs and western blots show a significant decrease in the mRNA and protein expression levels, respectively, of the glioblastoma stem cell (GSC) markers CD133, SOX2, and Nestin in the GB5 RAP2A-expressing neurospheres (GB5-RAP2A) compared to the control neurospheres (GB5). Data shown in the scaled bar graphs was analyzed with an unpaired two-tailed Student's *t*-test; error bars show the SD; n=3 different experiments; *p<0.05, **p<0.01.

The online version of this article includes the following source data for figure 4:

**Source data 1.** Original files for the blots displayed in *Figure 4*.

**Source data 2.** Original files for the blots displayed in *Figure 4*, labeling the relevant bands.

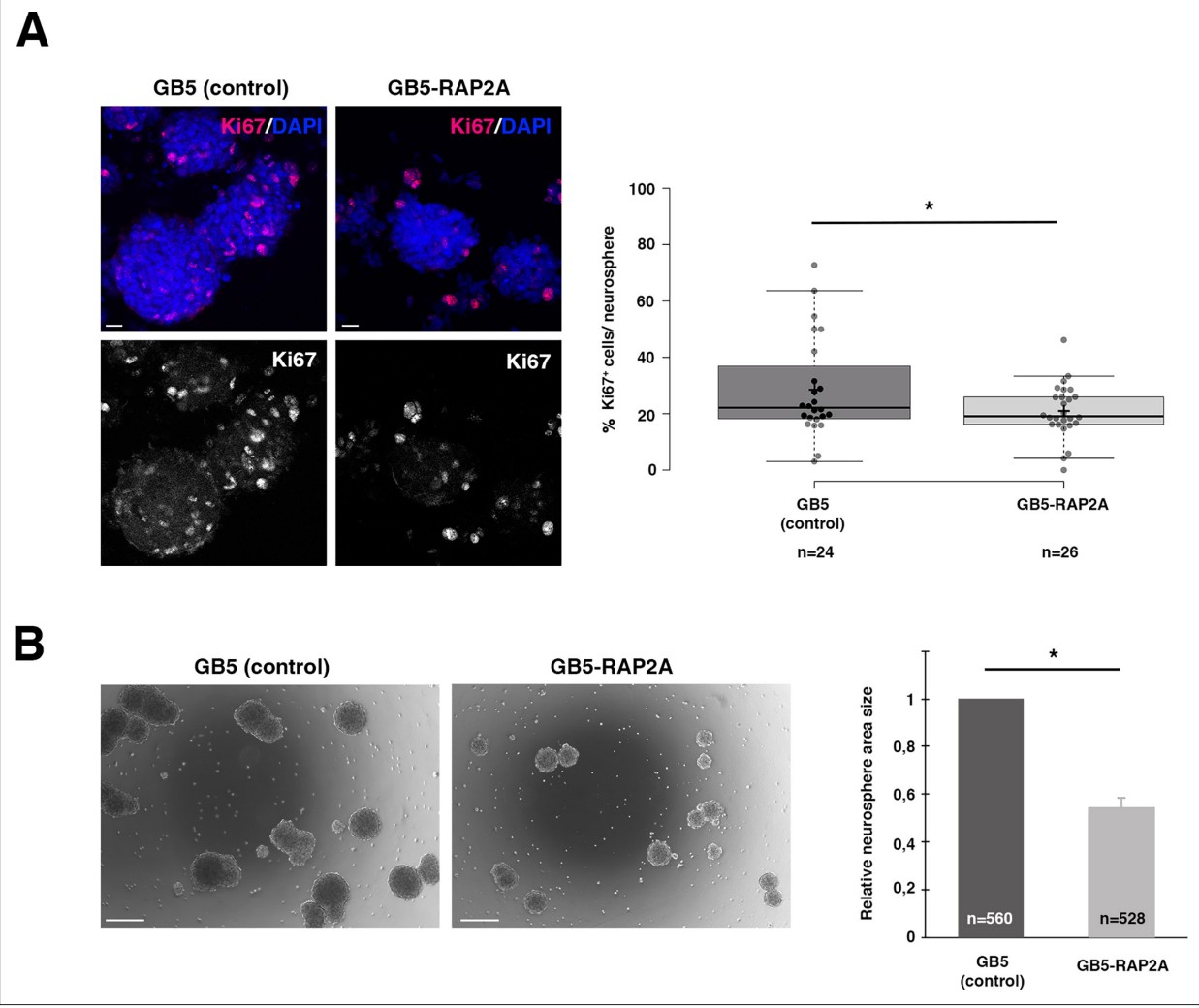

**Figure 5.** *RAP2A* expression in glioblastoma (GBM) neurosphere cultures decreases cell proliferation and sphere size. (**A**) GB5 neurospheres expressing RAP2A (GB5-RAP2A) show a significantly lower number of Ki67-expressing cells per neurosphere than control GB5 neurospheres. Data shown in the box plots was analyzed with a *t*-test; the central lines represent the median and the box limits the lower and upper quartiles, as determined using R software; crosses represent sample means; error bars indicate the SEM; n=number of sample points. (**B**) GB5 neurospheres expressing RAP2A (GB5-RAP2A) show a significant decrease in their size compared to control GB5 neurospheres of the same stage. Data shown in the scaled bar graphs was analyzed with an unpaired two-tailed Student's *t*-test; error bars show the SD; n=total number of neurospheres of three different experiments; *p<0.05; scale bars: 10 µm (in **A**) and 100 µm (in **B**).

The online version of this article includes the following source data for figure 5:

**Source data 1.** Source data of *Figure 5A* analyisis.

**Source data 2.** Source data of *Figure 5B* analysis.

a first approach, we aimed to analyze whether the levels of human homologs of known *Drosophila* ACD regulators were altered in human tumors. Specifically, we centered on human GBM, as the presence of CSCs (GSCs) has been shown in this tumor. The microarray we interrogated with GBM patient samples had some limitations. For example, not all the human gene homologs of the *Drosophila* ACD regulators were present (i.e., the human homologs of the determinant Numb). Likewise, we only tested seven different GBM patient samples. Nevertheless, the output from this analysis was enough to determine that most of the human genes tested in the array presented altered levels of expression. We selected for further analyses *RAP2A,* one of the human genes that showed the lowest levels of expression compared to the control samples. However, it would be interesting to analyze in the future the potential consequences that altered levels of expression of the other human homologs in the array can have in the behavior of the GSCs. In silico analyses, taking advantage of the existence

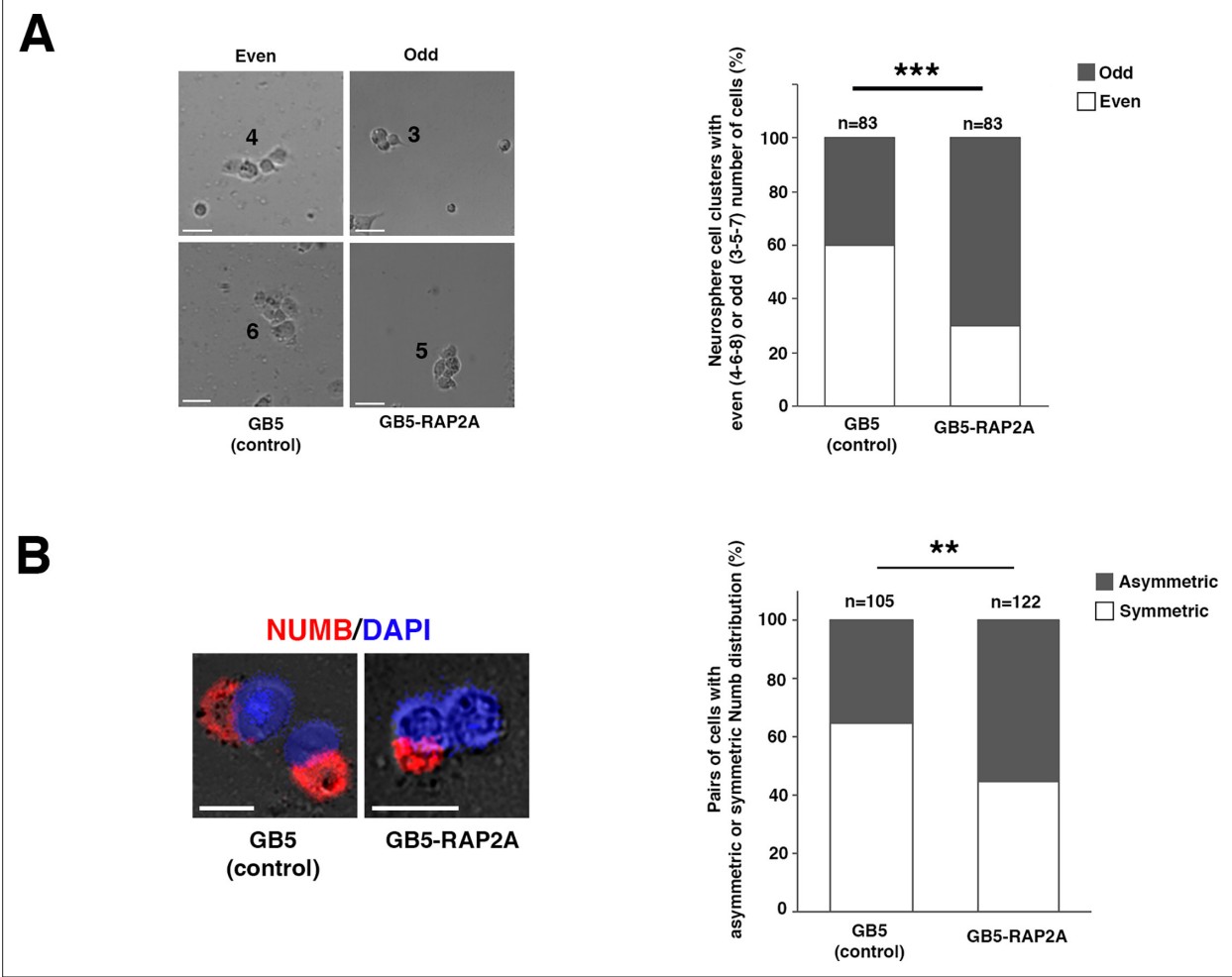

**Figure 6.** *RAP2A* expression in glioblastoma (GBM) neurosphere cultures fosters asymmetric cell division (ACD) in glioblastoma stem cells (GSCs). (**A**) Early-stage GB5 neurospheres expressing RAP2A (GB5-RAP2A) show an odd number of cells (3-5-7) significantly more frequently than control GB5 neurospheres, which show more frequently an even number of cells (4-6-8). (**B**) GB5-RAP2A dividing cells show a significant increase in the number of asymmetric NUMB localization in the progeny compared to control GB5 dividing cells. Data shown in the bar graphs was analyzed with a chi-square test with Yates correction; n=number of neurosphere cell clusters analyzed. **p<0.01; ***p<0.001, scale bar: 20 μm.

The online version of this article includes the following source data for figure 6:

**Source data 1.** Source data of *Figure 6A* analysis.

**Source data 2.** Source data of *Figure 6B* analysis.

of established datasets, such as the TCGA, can help to more robustly assess, in a bigger sample size, the relevance of those human genes' expression levels in GBM progression, as we observed for the gene *RAP2A*.

We have previously shown that *Drosophila* Rap1 acts as a novel NB ACD regulator in a complex with other small GTPases and the apical regulators Canoe (Cno), aPKC, and Par-6 (*Carmena et al., 2011*). Here, we have shown that *Drosophila* Rap2l also regulates NB ACD by ensuring the correct localization of the ACD modulators Cno and Numb. We have also demonstrated that RAP2A, the human homolog of *Drosophila* Rap2l, behaves as an ACD regulator in GBM neurosphere cultures, and its restitution to these GBM cultures, in which it is present at low levels, targets the stemness of GSCs, increasing the number of ACDs. It would be of great interest in the future to determine the specific mechanism by which Rap2l/RAP2A is regulating this process. One possibility is that, as it occurs in the case of the *Drosophila* ACD regulator Rap1, Rap2l/RAP2A is physically interacting or in a complex with other relevant ACD modulators. Thus, this study supports the relevance of ACD in CSCs of human tumors to refrain the expansion of the tumor. Other studies, however, claim that ACD

should be targeted in human tumors as it promotes the intratumoral heterogeneity that hampers the complete tumor loss after chemotherapy (*Samanta et al., 2023*; *Chao et al., 2023*; *Hitomi et al., 2021*). More investigations should be carried out with other human gene homologs of ACD regulators to further confirm the results of this study. Likewise, analyses in vivo (i.e., in mouse xenografts) would also be required to reinforce our conclusions. This would be very relevant in order to consider ACD restitution in CSCs of human tumors, what has been called 'differentiation therapy' (*de Thé, 2018*), as a potential alternative therapeutic treatment.

## Materials and methods

### *Drosophila* husbandry, strains, and genetics

The following fly stocks were used: *wor-Gal4 ase-Gal80* (*Neumüller et al., 2011*); *UAS-CD8::GFP* (Bloomington Drosophila Stock Center [BDSC] #5137); *UAS-CD2* (BDSC #1373); *UAS-Rap2l*^*KKRNAi*^ (Vienna Drosophila Resource Center [VDRC] #107745); *UAS-Rap2l*^*GDRNAi*^ (VDRC #45228); *UAS-Rap2l*^*BDSCRNAi*^ (BDSC #51840); *cno::GFP* (*Rives-Quinto et al., 2017*). All the fly stocks were raised and kept at 18°C or 25°C incubators. Experimental temperatures for the assays were maintained using 25°C or 29°C incubators. The *Gal-4 × UAS*-crosses were carried out at 29°C (the first 2 days, they were kept at 25°C, and then transferred to a new tube and left at 29°C) until third-instar larvae developed.

### *Drosophila* histology, immunofluorescence, and microscopy

Larval brains were dissected out in PBS (Phosphate-Buffered Saline) and fixed with 4% PFA (Paraformaldehyde) in PBT (PBS and Triton X-100 0.1%) for 20 min at room temperature with gentle rocking. Fixed brains were washed three times for 15 min with PBT (PBS and Triton X-100 0.3%) and then incubated in PBT-BSA for at least 1 h before incubation with the corresponding primary antibody/antibodies. The following primary antibodies were used in this study: guinea pig anti-Dpn (1:2000) (*Rives-Quinto et al., 2017*), rabbit anti-Ase (1:100) (*Rives-Quinto et al., 2017*), rabbit anti-PKC ζ (1:100; Santa Cruz Biotechnology, sc-216, RRID:AB_2300359), goat anti-Numb (1:200; Santa Cruz Biotechnology, sc-23579, RRID:AB_653503), mouse anti-PH3 (1:2000; Millipore, 05-806, RRID:AB_310016), mouse anti-γTub (1:200; Sigma-Aldrich, T5326, RRID:AB_532292), and mouse anti-CD2 (1:50; Bio-Rad, RRID:AB_207382). Secondary antibodies conjugated to fluorescent dyes (Invitrogen) were used at 1:400 dilution. Samples were mounted in VECTASHIELD antifade mounting medium for fluorescence (Vector Laboratories, H-1000). Fluorescent images were captured using a Super-resolution Inverted Confocal Microscope Zeiss LSM 880-Airyscan Elyra PS.1. Images were analyzed using the image processing package Fiji from ImageJ and assembled using Adobe Photoshop CS6.

### GBM microarray

The cohort of patients consisted of seven GBM biopsies obtained from the Valencian Network of Biobanks. The procedures for obtaining tissue samples were developed in accordance with national ethical and legal standards and following the guidelines established in the Declaration of Helsinki, and approved by the Clinical Research Ethics Committee of the Hospital General Universitario de Elche, Spain. All patients had been diagnosed with GBM grade IV, and all of them exhibited an *IDH1* plus *IDH2* wild-type genotype. Normal human brain RNA (frontal lobe; pool from five donors) was purchased from BioChain and used as a control for gene expression experiments. A total of 12,516 genes were used to generate a custom Agilent Two-Color 8 × 15 K Agilent gene expression DNA microarray, which were deposited in the NCBI Gene Expression Omnibus database under accession number GSE182697 (*Larriba et al., 2024*).

### Bioinformatics analysis

The level of expression of human homologs of *Drosophila* ACD regulators was assessed in seven different GBM patients versus control samples of the custom-made microarray (GSE182697) (*Larriba et al., 2024*). Statistical analysis of microarray data was done in Multiexperiment Viewer (MeV) (*Saeed et al., 2003*). Hierarchical clustering analyses were done by optimizing gene and sample leaf order and using Pearson correlation for the distance metric selection. Average linkage clustering was used as a linkage method. For the GDM, distance, inverse of similarity, between two genes was calculated using a distance metric as indicated in the MeV.

## Human samples and cell lines

Glioma samples were obtained with patients' written informed consent in accordance with the Declaration of Helsinki and approved by the Ethics Committee at Hospital 12 de Octubre, Madrid, Spain (CEI 14/023). The primary GBM cell lines were provided by the Biobank at 'Hospital Ramón y Cajal' and 'Hospital 12 de Octubre'. Fresh tissue samples were enzymatically dissociated using Accumax (Millipore) and cultured in stem cell medium (Neurobasal, Invitrogen), supplemented with N2 (1:100, Invitrogen), GlutaMAX (1:100, Invitrogen), penicillin–streptomycin (1:100, Lonza), 0.4% heparin (Sigma-Aldrich), 40 ng/ml EGF, and 20 ng/ml bFGF2 (PeproTech). The GB cell lines used in this study were GB4, GB5, GB7, GB18, GB19, 12o89, Astros (*Gargini et al., 2016*; *Gargini et al., 2020*; see also the following table), and HEK293T (ATCC; RRID:CVCL_0063; identity authenticated by an STR profiling). The cell lines HEK293T and Astros were maintained in Dulbecco's modified Eagle's medium (DMEM) supplemented with 10% fetal bovine serum 2 mM L-glutamine, 0.1% penicillin (100 U/ml) and streptomycin (100 µg/ml), and non-essential amino acids (1×). All the cell lines tested negative for mycoplasma contamination.

| Primary cell line | Origin | Species | Sex | EGFR amp | EGFR mut | TP53 mut |
|---|---|---|---|---|---|---|
| GB4 | Hospital Ramón y Cajal (Madrid, Spain) | Human | Male | 1 | nd | 1 |
| GB5 | Hospital Ramón y Cajal | Human | Female | nd | nd | 1 |
| GB7 | Hospital Ramón y Cajal | Human | Male | nd | 1 | nd |
| GB18 | Hospital Ramón y Cajal | Human | Male | nd | 1 (vII) | 1 |
| GB19 | Hospital Ramón y Cajal | Human | Female | nd | nd | nd |
| 12o89 | Hospital 12 de Octubre (Madrid, Spain) | Human | Male | | 1 (V774M) | 1 |
| Astros50 | ScienCell Research Laboratories | Human | nd | wt | wt | wt |
| Astros 60 | ScienCell Research Laboratories | Human | nd | wt | wt | wt |

GBM cell lines: nd, not diagnosed; 1, altered; wt, wild-type.

## GBM-derived neurosphere cell cultures

For stem cell culture of GBM-derived neuroespheres, we used DMEM:F12 with FGF-2 (20 ng/ml), EGF-2 (20 ng/ml), and N2 supplements, 0.1% penicillin (100 U/ml), and streptomycin (100 µg/ml), as previously described (*Ponti et al., 2005*; *Gargini et al., 2015*). The tumor spheres were dissociated to single cells using Accumax, and $1 \times 10^5$ dissociated cells/ml were plated in a flask and cultured for 6–7 days.

## In silico analysis

Data from TCGA and the Chinese Glioma Genome Atlas for GBM cohorts was accessed through the UCSC Xena-Browser (https://xenabrowser.net/) and Gliovis (https://gliovis.bioinfo.cnio.es/) for extraction of OS data, gene expression levels, and distribution of genetic alterations. Kaplan–Meier survival curves were generated by stratifying samples into low- and high-expression groups for each gene. Survival differences between these groups were assessed using the log-rank test.

## DNA constructs and lentiviral production

Lentiviral vectors were used to produce cells overexpressing *RAP2A* (*pLJM1RAP2A* Plasmid #19311, Addgene) or eGFP (pLJMEGFP Plasmid #19319, Addgene) as infection control. Infected cells were selected with puromycin. To obtain the lentivirus, HEK293T cells were transiently co-transfected with 5 µg of the appropriate lentiviral plasmid (pLJEGFP or pLJRRap2A, respectively), 5 µg of the packaging plasmid pCMVdR8.74 (Plasmid #22036, Addgene), and 5 µg of VERSUSV-G envelope protein plasmid pMD2G (Plasmid #12259, Addgene) using Lipofectamine Plus reagent (Invitrogen, Carlsbad, CA, USA). Lentiviral supernatants were collected after 48 h of HEK293T transfection.

## Neurosphere immunofluorescences and imaging

For immunostaining, floating cultured neurospheres were incubated on Matrigel-coated microslide glass for 2 h at 37°C. After having been washed, the attached neurospheres were fixed in 4%

paraformaldehyde for 20 min. Cells were blocked for 1 h in 1% FBS (Fetal Bovine Serum) and 0.1% Triton X-100 in PBS; then, they were incubated for 2 h with the primary antibody: mouse anti-CD133 (HB#7, Developmental Studies Hybridoma Bank, RRID:AB_2619580), mouse anti-Ki67 (3E6, Developmental Studies Hybridoma Bank, RRID:AB_2617702), rabbit anti-Sox2 (N1C3, GTX101507 GeneTex, RRID:AB_2038021), mouse anti-Nestin (10c2, sc-23927, Santa Cruz Biotechnology, RRID:AB_627994), and mouse anti-Numb (48, sc-136554, Santa Cruz Biotechnology, RRID:AB_10611794). The secondary antibodies used were Alexa555 goat anti-rabbit (A32732, Invitrogen) or Alexa555 goat anti-mouse (A32727, Invitrogen). Phalloidin 633 (Sigma-Aldrich, 68825 Phalloidin-Atto 633) was added and incubated for 20 min in PBS; VECTASHIELD mounting medium with DAPI (Linaris, H-1200) was added before mounting. Fluorescent images were captured using a Super-resolution Inverted Confocal Microscope Zeiss LSM 880-Airyscan Elyra PS.1. Images were analyzed using the image processing package Fiji from ImageJ and assembled using Adobe Photoshop CS6 program.

## Immunofluorescent quantification
Fluorescence intensity measurements were carried out using ImageJ software package. In all cases, images of neuroespheres transfected with GFP (GB5, control) or RAP2A (GB5-RAP2A) lentivirus were analyzed.

### CD133
Using the 'line tool', a line was drawn on the channel with the CD133 signal on three different pictures per genotype. Fluorescence was measured across the line sections with the 'Plot Profile Tool' and reported as a subset of images showing the signal intensity plot. For quantification, only the 100 highest gray value points were taken.

### Sox
After adjusting the threshold on the channel with the Sox2 signal and eliminating particles smaller than 5 microns, the fluorescence intensity was measured as the mean gray value (sum of all pixels with gray values inside the selection, divided by the number of pixels). *Figure 3* shows one example out of three quantifications achieved with at least 20 cells per genotype.

### Nestin
The regions of interest (ROIs) were set with the 'hand drawing tool' using the Phalloidin staining to follow the cell periphery. At least 20 cells (ROIs) from three different pictures were selected per condition. The background fluorescence was subtracted from the mean gray value fluorescence for each well, and the fluorescence intensity was normalized by the nuclear DAPI signal coming from the same cell (ROI). The plot in *Figure 3C* represents the mean gray value, that is, the sum of all pixels with gray values inside the selection divided by the number of pixels.

## RNA isolation and qRT-PCR assay
Total RNA was extracted from 7-day-old neurospheres using TRI reagent (AM9738, Invitrogen) following manufacturer's indications and quantified using a nanodrop (ND-1000, Thermo Scientific). 2 ug of RNA was treated with DNaseI (EN0521, Thermo Scientific) and reverse transcribed with SuperScript III Reverse Transcriptase (1808044, Invitrogen). Quantitative real-time PCR (qRT-PCR) was performed using Power SYBR Green PCR Master Mix (PN4367218, Applied Biosystems), following established protocols with 60°C for annealing/extension and 40 cycles of amplification on a QuantStudio 3 apparatus (Applied Biosystems). Act5 primers were used for normalization, and relative expression was calculated using the comparative ΔΔCt. The average of at least three independent experiments is shown in *Figure 4*.

## Neurosphere western blots
Protein extracts were prepared by resuspending cell pellets in a lysis buffer (50 mM Tris, pH 7.5, 300 mM NaCl, 0.5% SDS (Sodium Dodecyl Sulfate), and 1% Triton X-100) and incubating the cells for 10 min at 95°C. The lysed extracts were centrifuged at 13,000 × $g$ for 10 min at room temperature, and the protein concentration was determined using a commercially available colorimetric assay (BCA Protein Assay Kit). Approximately 25 µg of protein was resolved by 10% or 12,5% SDS-PAGE, and

they were then transferred to a PVDF (Polyvinylidene Fluoride) membrane (Hybond-ECL, Amersham Biosciences). The membranes were blocked for 1 h at room temperature in TBS-T (10 mM Tris–HCl, pH 7.5, 100 mM NaCl, and 0.1% Tween-20) with 5% skimmed milk and then incubated overnight at 4°C, with the corresponding primary antibody diluted in TBS-T with 2.5% skimmed milk. After washing three times with TBS-T, the membranes were incubated for 2 h at room temperature, with their corresponding secondary antibody diluted in TBS-T with 2.5% skimmed milk. The proteins were visible by enhanced chemiluminescence with ECL (Pierce) using Amersham imager 680, and the signal was quantified using Fiji-ImageJ software.

### Proliferative index (Ki67 analysis)

The number of Ki67+ cells with respect to the total number of nuclei labeled with DAPI within a given neurosphere was counted to calculate the proliferative index, which was expressed as the percentage of Ki67+ cells over total DAPI+ cells. More than 20 neurospheres coming from different pictures were analyzed.

### Sphere size measurement

GBM cells (GB5 cell line) were infected by control lentivirus (LV-GFP) or lentivirus directing expression of *RAP2A* (LV-*RAP2A*). The tumor spheres were Accumax-dissociated to single cells during three consecutive passages. Seven days after the third plating in a six-well plate, spheres were captured using a ×10 objective (Carl Zeiss Axio Imager A1 microscope). Twenty random pictures of each genotype were taken, and the images were analyzed by measuring the area of all the spheres present using the 'freeform drawing tool' in the ImageJ software package. The data presented shows the average area measured in three independent experiments.

### Pair assay and Numb segregation analysis

Dissociated cells were plated at a low density (50,000 cells/ml) on Matrigel-coated well glass chamber slides on DMEM:F12 with FGF-2 (20 ng/ml), EGF-2 (20 ng/ml), and N2 supplements, 0.1% penicillin (100 U/ml), and streptomycin (100 µg/ml), supplemented with puromycin. Cultures were incubated for 24, 48, or 72 h at 37°C with 5% $CO_2$. Bright-field microscopy images from each time point were taken, and then the slides were fixed and immunolabeled for Numb, as well as for DAPI, to identify nuclei. First, isolated groups of cells (most probably coming from the same single progenitor) were identified, and the number of cells in each cluster was annotated. Groups of 3–5 cells were counted in the slide incubated for 48 h and groups of 6–8 cells in that incubated for 72 h. We did not include in the analysis 'neurospheres' with two cells, as these can be the result of a symmetric or an asymmetric cell division. We took into account neurospheres with 3–5–7 cells (as odd number of cells) and 4–6–8 cells (as even number of cells).

For the Numb analysis, the localization of Numb was visualized and scored in at least 100 DAPI-labeled cell pairs. To avoid any artifact, pairs were only captured in the slide incubated for 24 h. The images in *Figure 6* were acquired using an Inverted Microscope Leica Thunder Imager.

### Statistical analysis

The data were first analyzed using the Shapiro–Wilk test to determine whether the sample followed a normal distribution. Parametric *t*-test or a nonparametric two-tailed Mann–Whitney *U* test for those that did not follow a normal distribution was used to compare statistical differences between two different groups. Mean and standard deviation values were calculated by standard methods. A chi-square test was used to analyze the data in *Figure 2C*. A Kruskal–Wallis test followed by post hoc Dunn's test was used to analyze the data in *Figure 2—figure supplement 1*. To assume statistical significance, p-values were determined to be <0.05. Sample size (n) and p-values are indicated in the figure or figure legends; *p<0.05, **p<0.01, ***p<0.001, ****p<0.0001; ns=not significant (p>0.05).

## Acknowledgements

We thank the Bloomington Drosophila Stock Center at the University of Indiana and the Developmental Studies Hybridoma Bank at the University of Iowa for kindly providing fly strains and reagents. RG was funded by 'P through the project Miguel Servet Contracts' (CP21/00116), PI22/01171, and co-funded by the European Union. MS was supported by the Spanish grant from

the Instituto de Salud Carlos III (ISCIII) PI22/00824. AC was financed by the Spanish grants from the Ministry of Science, Innovation and Universities (PGC2018-097093-B-100), the Ministry of Science and Innovation (PID2021-123196NB-100), the Generalitat Valenciana Prometeo/2021/052 grant, and the European Regional Development Fund (FEDER). The 'Instituto de Neurociencias' in Alicante is a 'Severo Ochoa Center of Excellence in R&D'. Our group belongs to 'Conexión Cáncer (CSIC)'.

## Additional information

### Funding

| Funder | Grant reference number | Author |
|---|---|---|
| Instituto de Salud Carlos III | PI22/01171 | Ricardo Gargini |
| Instituto de Salud Carlos III | PI22/00824 | Miguel Saceda |
| Ministerio de Ciencia, Innovación y Universidades | PGC2018-097093-B-100 | Ana Carmena |
| Ministerio de Ciencia e Innovación | PID2021-123196NB-100 | Ana Carmena |
| Generalitat Valenciana | Prometeo/2021/052 | Ana Carmena |
| European Regional Development Fund | | Ricardo Gargini Ana Carmena |

The funders had no role in study design, data collection and interpretation, or the decision to submit the work for publication.

### Author contributions

Maribel Franco, participated in the design of the experiments, conducted most of them and analyzed the data; Ricardo Gargini, developed the GB lines and performed the in silico analyses; Víctor M Barberá, carried out the GB microarray and bioinformatics analysis; Daniel Becerra, completed some experiments (Figure 2-figure supplement 1); Miguel Saceda, carried out the GB microarray and bioinformatics analysis; Ana Carmena, conceived the study, participated in the design of the experiments and wrote the manuscript

### Author ORCIDs
Daniel Becerra ![ORCID] https://orcid.org/0009-0008-9076-9695
Ana Carmena ![ORCID] https://orcid.org/0000-0003-1855-7934

Reviewer #1 (Public review): https://doi.org/10.7554/eLife.105690.3.sa1
Reviewer #2 (Public review): https://doi.org/10.7554/eLife.105690.3.sa2
Author response https://doi.org/10.7554/eLife.105690.3.sa3

## Additional files

### Supplementary files
MDAR checklist

### Data availability
All data generated or analyzed during this study are included in the manuscript and supporting files; source data files have been provided.

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
