## [Editor Report · eLife Assessment]

This **useful** study explores the role of *RAP2A* in asymmetric cell division (ACD) regulation in glioblastoma stem cells (GSCs), drawing parallels to *Drosophila* ACD mechanisms and proposing that an imbalance toward symmetric divisions drives tumor progression. While findings on *RAP2A*’s role in GSC expansion are promising, and the reviewers found the study innovative and technically **solid**, the study relies on neurosphere models without in vivo confirmation and will therefore need to be further validated in the future.

---

## [Referee Report · Reviewer #1 (Public review)]

Summary:

The authors validate the contribution of RAP2A to GB progression. RAp2A participates in asymetric cell division, and the localization of several cell polarity markers including cno and Numb.

Strengths:

The use of human data, *Drosophila* models and cell culture or neurospheres is a good scenario to validate the hypothesis using complementary systems.

Moreover, the mechanisms that determine GB progression, and in particular glioma stem cells biology, are relevant for the knowledge on glioblastoma and opens new possibilities to future clinical strategies.

Weaknesses:

While the manuscript presents a well-supported investigation into RAP2A's role in GBM, some methodological aspects could benefit from further validation. The major concern is the reliance on a single GB cell line (GB5), including multiple GBM lines, particularly primary patient-derived 3D cultures with known stem-like properties, would significantly enhance the study's robustness.

Several specific points raised in previous reviews have improved this version of the manuscript:

• The specificity of Rap2l RNAi has been further confirmed by using several different RNAi tools.

• Quantification of phenotypic penetrance and survival rates in Rap2l mutants would help determine the consistency of ACD defects. The authors have substantially increased the number of samples analyzed including three different RNAi lines (both the number of NB lineages and the number of different brains analyzed) to confirm the high penetrance of the phenotype.

• The observations on neurosphere size and Ki-67 expression require normalization (e.g., Ki-67+ cells per total cell number or per neurosphere size). This is included in the manuscript and now clarified in the text.

• The discrepancy in Figures 6A and 6B requires further discussion. The authors have included a new analysis and further explanations and they can conclude that in 2 cell-neurospheres there are more cases of asymmetric divisions in the experimental condition (RAP2A) than in the control.

• Live imaging of ACD events would provide more direct evidence. Live imaging was not done due to technical limitations. Despite being a potential contribution to the manuscript, the current conclusions of the manuscript are supported by the current data, and live experiments can be dispensable

• Clarification of terminology and statistical markers (e.g., p-values) in Figure 1A would improve clarity. This has been improved.

Comments on revisions:

The manuscript has improved the clarity in general, and I think that it is suitable for publication. However, for future experiments and projects, I would like to insist in the relevance of validating the results in vivo using xenografts with 3D-primary patient-derived cell lines or GB organoids.

---

## [Referee Report · Reviewer #2 (Public review)]

This study investigates the role of RAP2A in regulating asymmetric cell division (ACD) in glioblastoma stem cells (GSCs), bridging insights from *Drosophila* ACD mechanisms to human tumor biology. They focus on RAP2A, a human homolog of *Drosophila* Rap2l, as a novel ACD regulator in GBM is innovative, given its underexplored role in cancer stem cells (CSCs). The hypothesis that ACD imbalance (favoring symmetric divisions) drives GSC expansion and tumor progression introduces a fresh perspective on differentiation therapy. However, the dual role of ACD in tumor heterogeneity (potentially aiding therapy resistance) requires deeper discussion to clarify the study's unique contributions against existing controversies.

Comments on revisions:

More experiments as suggested in the original assessment of the submission are needed to justify the hypothesis drawn in the manuscript.

---

## [Author Response]

The following is the authors’ response to the original reviews.

**Reviewer #1 (Public review):**
Summary:The authors validate the contribution of RAP2A to GB progression. RAp2A participates in asymmetric cell division, and the localization of several cell polarity markers, including cno and Numb.Strengths:The use of human data, *Drosophila* models, and cell culture or neurospheres is a good scenario to validate the hypothesis using complementary systems.Moreover, the mechanisms that determine GB progression, and in particular glioma stem cells biology, are relevant for the knowledge on glioblastoma and opens new possibilities to future clinical strategies.Weaknesses:While the manuscript presents a well-supported investigation into RAP2A's role in GBM, several methodological aspects require further validation. The major concern is the reliance on a single GB cell line (GB5), which limits the generalizability of the findings. Including multiple GBM lines, particularly primary patient-derived 3D cultures with known stem-like properties, would significantly enhance the study's relevance.Additionally, key mechanistic aspects remain underexplored. Further investigation into the conservation of the Rap2l-Cno/aPKC pathway in human cells through rescue experiments or protein interaction assays would be beneficial. Similarly, live imaging or lineage tracing would provide more direct evidence of ACD frequency, complementing the current indirect metrics (odd/even cell clusters, Numb asymmetry).Several specific points require attention:(1) The specificity of Rap2l RNAi needs further confirmation. Is Rap2l expressed in neuroblasts or intermediate neural progenitors? Can alternative validation methods be employed?

There are no available antibodies/tools to determine whether Rap2l is expressed in NB lineages, and we have not been able either to develop any. However, to further prove the specificity of the Rap2l phenotype, we have now analyzed two additional and independent RNAi lines of Rap2l along with the original RNAi line analyzed. We have validated the results observed with this line and found a similar phenotype in the two additional RNAi lines now analyzed. These results have been added to the text ("Results section", page 6, lines 142-148) and are shown in Supplementary Figure 3.

(2) Quantification of phenotypic penetrance and survival rates in Rap2l mutants would help determine the consistency of ACD defects.

In the experiment previously mentioned (repetition of the original Rap2l RNAi line analysis along with two additional Rap2l RNAi lines) we have substantially increased the number of samples analyzed (both the number of NB lineages and the number of different brains analyzed). With that, we have been able to determine that the penetrance of the phenotype was 100% or almost 100% in the 3 different RNAi lines analyzed (n>14 different brains/larvae analyzed in all cases). Details are shown in the text (page 6, lines 142-148), in Supplementary Figure 3 and in the corresponding figure legend.

(3) The observations on neurosphere size and Ki-67 expression require normalization (e.g., Ki-67+ cells per total cell number or per neurosphere size). Additionally, apoptosis should be assessed using Annexin V or TUNEL assays.

The experiment of Ki-67+ cells was done considering the % of Ki-67+ cells respect the total cell number in each neurosphere. In the "Materials and methods" section it is well indicated: "The number of Ki67+ cells with respect to the total number of nuclei labelled with DAPI within a given neurosphere were counted to calculate the Proliferative Index (PI), which was expressed as the % of Ki67+ cells over total DAPI+ cells"

Perhaps it was not clearly showed in the graph of Figure 5A. We have now changed it indicating: "% of Ki67+ cells/ neurosphere" in the "Y axis".

Unfortunately, we currently cannot carry out neurosphere cultures to address the apoptosis experiments.

(4) The discrepancy in Figures 6A and 6B requires further discussion.

We agree that those pictures can lead to confusion. In the analysis of the "% of neurospheres with even or odd number of cells", we included the neurospheres with 2 cells both in the control and in the experimental condition (RAP2A). The number of this "2 cell-neurospheres" was very similar in both conditions (27,7 % and 27 % of the total neurospheres analyzed in each condition), and they can be the result of a previous symmetric or asymmetric division, we cannot distinguish that (only when they are stained with Numb, for example, as shown in Figure 6B). As a consequence, in both the control and in the experimental condition, these 2-cell neurospheres included in the group of "even" (Figure 6A) can represent symmetric or asymmetric divisions. However, in the experiment shown in Figure 6B, it is shown that in these 2 cellneurospheres there are more cases of asymmetric divisions in the experimental condition (RAP2A) than in the control.

Nevertheless, to make more accurate and clearer the conclusions, we have reanalyzed the data taking into account only the neurospheres with 3-5-7 (as odd) or 4-6-8 (as even) cells. Likewise, we have now added further clarifications regarding the way the experiment has been analyzed in the methods.

(5) Live imaging of ACD events would provide more direct evidence.

We agree that live imaging would provide further evidence. Unfortunately, we currently cannot carry out neurosphere cultures to approach those experiments.

(6) Clarification of terminology and statistical markers (e.g., p-values) in Figure 1A would improve clarity.

We thank the reviewer for pointing out this issue. To improve clarity, we have now included a Supplementary Figure (Fig. S1) with the statistical parameters used. Additionally, we have performed a hierarchical clustering of genes showing significant or not-significant changes in their expression levels.

(7) Given the group's expertise, an alternative to mouse xenografts could be a *Drosophila* genetic model of glioblastoma, which would provide an in vivo validation system aligned with their research approach.

The established *Drosophila* genetic model of glioblastoma is an excellent model system to get deep insight into different aspects of human GBM. However, the main aim of our study was to determine whether an imbalance in the mode of stem cell division, favoring symmetric divisions, could contribute to the expansion of the tumor. We chose human GBM cell lines-derived neurospheres because in human GBM it has been demonstrated the existence of cancer stem cells (glioblastoma or glioma stem cells -GSCs--). And these GSCs, as all stem cells, can divide symmetric or asymmetrically. In the case of the *Drosophila* model of GBM, the neoplastic transformation observed after overexpressing the EGF receptor and PI3K signaling is due to the activation of downstream genes that promote cell cycle progression and inhibit cell cycle exit. It has also been suggested that the neoplastic cells in this model come from committed glial progenitors, not from stem-like cells.

With all, it would be difficult to conclude the causes of the potential effects of manipulating the Rap2l levels in this *Drosophila* system of GBM. We do not discard this analysis in the future (we have all the "set up" in the lab). However, this would probably imply a new project to comprehensively analyze and understand the mechanism by which Rap2l (and other ACD regulators) might be acting in this context, if it is having any effect.

However, as we mentioned in the Discussion, we agree that the results we have obtained in this study must be definitely validated in vivo in the future using xenografts with 3D-primary patient-derived cell lines.

**Reviewer #2 (Public review):**
This study investigates the role of RAP2A in regulating asymmetric cell division (ACD) in glioblastoma stem cells (GSCs), bridging insights from *Drosophila* ACD mechanisms to human tumor biology. They focus on RAP2A, a human homolog of *Drosophila* Rap2l, as a novel ACD regulator in GBM is innovative, given its underexplored role in cancer stem cells (CSCs). The hypothesis that ACD imbalance (favoring symmetric divisions) drives GSC expansion and tumor progression introduces a fresh perspective on differentiation therapy. However, the dual role of ACD in tumor heterogeneity (potentially aiding therapy resistance) requires deeper discussion to clarify the study's unique contributions against existing controversies. Some limitations and questions need to be addressed.(1) Validation of RAP2A's prognostic relevance using TCGA and Gravendeel cohorts strengthens clinical relevance. However, differential expression analysis across GBM subtypes (e.g., MES, DNA-methylation subtypes) should be included to confirm specificity.

We have now included a Supplementary figure (Supplementary Figure 2), in which we show the analysis of RAP2A levels in the different GBM subtypes (proneural, mesenchymal and classical) and their prognostic relevance (i.e. the proneural subtype that presents RAP2A levels significantly higher than the others is the subtype that also shows better prognostic).

(2) Rap2l knockdown-induced ACD defects (e.g., mislocalization of Cno/Numb) are well-designed. However, phenotypic penetrance and survival rates of Rap2l mutants should be quantified to confirm consistency.

We have now analyzed two additional and independent RNAi lines of Rap2l along with the original RNAi line. We have validated the results observed with this line and found a similar phenotype in the two additional RNAi lines now analyzed. To determine the phenotypic penetrance, we have substantially increased the number of samples analyzed (both the number of NB lineages and the number of different brains analyzed). With that, we have been able to determine that the penetrance of the phenotype was 100% or almost 100% in the 3 different Rap2l RNAi lines analyzed (n>14 different brains/larvae analyzed in all cases). These results have been added to the text ("Results section", page 6, lines 142-148) and are shown in Supplementary Figure 3 and in the corresponding figure legend.

(3) While GB5 cells were effectively used, justification for selecting this line (e.g., representativeness of GBM heterogeneity) is needed. Experiments in additional GBM lines (especially the addition of 3D primary patient-derived cell lines with known stem cell phenotype) would enhance generalizability.

We tried to explain this point in the paper (Results). As we mentioned, we tested six different GBM cell lines finding similar mRNA levels of RAP2A in all of them, and significantly lower levels than in control Astros (Fig. 3A). We decided to focus on the GBM cell line called GB5 as it grew well (better than the others) in neurosphere cell culture conditions, for further analyses. We agree that the addition of at least some of the analyses performed with the GB5 line using other lines (ideally in primary patientderive cell lines, as the reviewer mentions) would reinforce the results. Unfortunately, we cannot perform experiments in cell lines in the lab currently. We will consider all of this for future experiments.

(4) Indirect metrics (odd/even cell clusters, NUMB asymmetry) are suggestive but insufficient. Live imaging or lineage tracing would directly validate ACD frequency.

We agree that live imaging would provide further evidence. Unfortunately, we cannot approach those experiments in the lab currently.

(5) The initial microarray (n=7 GBM patients) is underpowered. While TCGA data mitigate this, the limitations of small cohorts should be explicitly addressed and need to be discussed.

We completely agree with this comment. We had available the microarray, so we used it as a first approach, just out of curiosity of knowing whether (and how) the levels of expression of those human homologs of *Drosophila* ACD regulators were affected in this small sample, just as starting point of the study. We were conscious of the limitations of this analysis and that is why we followed up the analysis in the datasets, on a bigger scale. We already mentioned the limitations of the array in the Discussion:

"The microarray we interrogated with GBM patient samples had some limitations. For example, not all the human genes homologs of the *Drosophila* ACD regulators were present (i.e. the human homologs of the determinant Numb). Likewise, we only tested seven different GBM patient samples. Nevertheless, the output from this analysis was enough to determine that most of the human genes tested in the array presented altered levels of expression"[....] In silico analyses, taking advantage of the existence of established datasets, such as the TCGA, can help to more robustly assess, in a bigger sample size, the relevance of those human genes expression levels in GBM progression, as we observed for the gene RAP2A."

(6) Conclusions rely heavily on neurosphere models. Xenograft experiments or patient-derived orthotopic models are critical to support translational relevance, and such basic research work needs to be included in journals.

We completely agree. As we already mentioned in the Discussion, the results we have obtained in this study must be definitely validated in vivo in the future using xenografts with 3D-primary patient-derived cell lines.

(7) How does RAP2A regulate NUMB asymmetry? Is the *Drosophila* Rap2l-Cno/aPKC pathway conserved? Rescue experiments (e.g., Cno/aPKC knockdown with RAP2A overexpression) or interaction assays (e.g., Co-IP) are needed to establish molecular mechanisms.

The mechanism by which RAP2A is regulating ACD is beyond the scope of this paper. We do not even know how Rap2l is acting in *Drosophila* to regulate ACD. In past years, we did analyze the function of another *Drosophila* small GTPase, Rap1 (homolog to human RAP1A) in ACD, and we determined the mechanism by which Rap1 was regulating ACD (including the localization of Numb): interacting physically with Cno and other small GTPases, such as Ral proteins, and in a complex with additional ACD regulators of the "apical complex" (aPKC and Par-6). Rap2l could be also interacting physically with the "Ras-association" domain of Cno (domain that binds small GTPases, such as Ras and Rap1). We have added some speculations regarding this subject in the Discussion:

"It would be of great interest in the future to determine the specific mechanism by which Rap2l/RAP2A is regulating this process. One possibility is that, as it occurs in the case of the *Drosophila* ACD regulator Rap1, Rap2l/RAP2A is physically interacting or in a complex with other relevant ACD modulators."

(8) Reduced stemness markers (CD133/SOX2/NESTIN) and proliferation (Ki-67) align with increased ACD. However, alternative explanations (e.g., differentiation or apoptosis) must be ruled out via GFAP/Tuj1 staining or Annexin V assays.

We agree with these possibilities. Regarding differentiation, the potential presence of increased differentiation markers would be in fact a logic consequence of an increase in ACD divisions/reduced stemness markers. Unfortunately, we cannot approach those experiments in the lab currently.

(9) The link between low RAP2A and poor prognosis should be validated in multivariate analyses to exclude confounding factors (e.g., age, treatment history).

We have now added this information in the "Results section" (page 5, lines 114-123).

(10) The broader ACD regulatory network in GBM (e.g., roles of other homologs like NUMB) and potential synergies/independence from known suppressors (e.g., TRIM3) warrant exploration.

The present study was designed as a "proof-of-concept" study to start analyzing the hypothesis that the expression levels of human homologs of known *Drosophila* ACD regulators might be relevant in human cancers that contain cancer stem cells, if those human homologs were also involved in modulating the mode of (cancer) stem cell division.

To extend the findings of this work to the whole ACD regulatory network would be the logic and ideal path to follow in the future.

We already mentioned this point in the Discussion:

"....it would be interesting to analyze in the future the potential consequences that altered levels of expression of the other human homologs in the array can have in the behavior of the GSCs. In silico analyses, taking advantage of the existence of established datasets, such as the TCGA, can help to more robustly assess, in a bigger sample size, the relevance of those human genes expression levels in GBM progression, as we observed for the gene RAP2A."

(11) The figures should be improved. Statistical significance markers (e.g., p-values) should be added to Figure 1A; timepoints/culture conditions should be clarified for Figure 6A.

Regarding the statistical significance markers, we have now included a Supplementary Figure (Fig. S1) with the statistical parameters used. Additionally, we have performed a hierarchical clustering of genes showing significant or notsignificant changes in their expression levels.

Regarding the experimental conditions corresponding to Figure 6A, those have now been added in more detail in "Materials and Methods" ("Pair assay and Numb segregation analysis" paragraph).

(12) Redundant *Drosophila* background in the Discussion should be condensed; terminology should be unified (e.g., "neurosphere" vs. "cell cluster").

As we did not mention much about *Drosophila* ACD and NBs in the "Introduction", we needed to explain in the "Discussion" at least some very basic concepts and information about this, especially for "non-drosophilists". We have reviewed the Discussion to maintain this information to the minimum necessary.

We have also reviewed the terminology that the Reviewer mentions and have unified it.

**Reviewer #1 (Recommendations for the authors):**
To improve the manuscript's impact and quality, I would recommend:(1) Expand Cell Line Validation: Include additional GBM cell lines, particularly primary patient-derived 3D cultures, to increase the robustness of the findings.(2) Mechanistic Exploration: Further examine the conservation of the Rap2lCno/aPKC pathway in human cells using rescue experiments or protein interaction assays.(3) Direct Evidence of ACD: Implement live imaging or lineage tracing approaches to strengthen conclusions on ACD frequency.(4) RNAi Specificity Validation: Clarify Rap2l RNAi specificity and its expression in neuroblasts or intermediate neural progenitors.(5) Quantitative Analysis: Improve quantification of neurosphere size, Ki-67 expression, and apoptosis to normalize findings.(6) Figure Clarifications: Address inconsistencies in Figures 6A and 6B and refine statistical markers in Figure 1A.(7) Alternative In Vivo Model: Consider leveraging a *Drosophila* glioblastoma model as a complementary in vivo validation approach.Addressing these points will significantly enhance the manuscript's translational relevance and overall contribution to the field.

We have been able to address points 4, 5 and 6. Others are either out of the scope of this work (2) or we do not have the possibility to carry them out at this moment in the lab (1, 3 and 7). However, we will complete these requests/recommendations in other future investigations.

**Reviewer #2 (Recommendations for the authors):**
Major Revision /insufficient required to address methodological and mechanistic gaps.(1) Enhance Clinical RelevanceValidate RAP2A's prognostic significance across multiple GBM subtypes (e.g., MES, DNA-methylation subtypes) using datasets like TCGA and Gravendeel to confirm specificity.Perform multivariate survival analyses to rule out confounding factors (e.g., patient age, treatment history).(2) Strengthen Mechanistic InsightsInvestigate whether the Rap2l-Cno/aPKC pathway is conserved in human GBM through rescue experiments (e.g., RAP2A overexpression with Cno/aPKC knockdown) or interaction assays (e.g., Co-IP).Use live-cell imaging or lineage tracing to directly validate ACD frequency instead of relying on indirect metrics (odd/even cell clusters, NUMB asymmetry).(3) Improve Model Systems & Experimental DesignJustify the selection of GB5 cells and include additional GBM cell lines, particularly 3D primary patient-derived cell models, to enhance generalizability.It is essential to perform xenograft or orthotopic patient-derived models to support translational relevance.(5) Address Alternative InterpretationsRule out other potential effects of RAP2A knockdown (e.g., differentiation or apoptosis) using GFAP/Tuj1 staining or Annexin V assays.Explore the broader ACD regulatory network in GBM, including interactions with NUMB and TRIM3, to contextualize findings within known tumor-suppressive pathways.(6) Improve Figures & ClarityAdd statistical significance markers (e.g., p-values) in Figure 1A and clarify timepoints/culture conditions for Figure 6A.Condense redundant *Drosophila* background in the discussion and ensure consistent terminology (e.g., "neurosphere" vs. "cell cluster").

We have been able to address points 1, partially 3 and 6. Others are either out of the scope of this work or we do not have the possibility to carry them out at this moment in the lab. However, we are very interested in completing these requests/recommendations and we will approach that type of experiments in other future investigations.